# Optimizing Phosphorus Fertilizer Use on the Loess Plateau: Impact on Soil Properties and Crop Production Efficiency

Chutao Liang [1], Xiaoqi Liu [1], Lei Feng [2], Ning Jin [3], Jialong Lv [1] and Qiang Yu [1,4,*]

[1] College of Natural Resources and Environment, Northwest A&F University, Yangling 712100, China; liangchutao17@nwafu.edu.cn (C.L.); liuxiaoqi2016@163.com (X.L.)

[2] College of Resource and Environment, Xinjiang Agricultural University, Urumqi 830000, China; 15199386380@163.com

[3] Department of Resources and Environmental Engineering, Shanxi Institute of Energy, Jinzhong 030600, China; jinn.13b@igsnrr.ac.cn

[4] State Key Laboratory of Soil Erosion and Dryland Farming on the Loess Plateau, Institute of Soil and Water Conservation, Northwest A&F University, Yangling 712100, China

\* Correspondence: yuq@nwafu.edu.cn

**Abstract:** Various phosphorus (P) fertilizers are commonly utilized in agricultural production on the Loess Plateau. However, there exists a widespread issue of improper matching between P fertilizers, crop types, and soil types. This study proposes a scientifically based approach to managing phosphate fertilizer through a matching experiment. A field experiment was conducted to investigate the effects of different P fertilizers on soil P profiles in a wheat–corn rotation between October 2017 and September 2021. The experiment adopted a randomized block design. P fertilizer was applied as a basal fertilizer at rates of 115 kg $P_2O_5$ ha$^{-1}$ during the wheat season and 90 kg $P_2O_5$ ha$^{-1}$ during the maize season. Nitrogen (N) fertilizer application rates were 120 kg N ha$^{-1}$ for wheat and 180 kg N ha$^{-1}$ for maize. N fertilizer was divided into two applications, with 60% applied at pre-planting and 40% at the jointing stage of wheat or the V12 stage of maize. P fertilizer variants utilized in the study included ammonium dihydrogen, ammonium phosphate, calcium-magnesia phosphate fertilizer, calcium superphosphate, and ammonium polyphosphate. The transformation process of phosphate was examined, revealing that the commonly considered dominant diammonium phosphate fertilizer was not the optimal choice in this production system. Ammonium polyphosphate, calcium superphosphate, and ammonium dihydrogen were deemed more suitable for application in Loess soil. Furthermore, an analysis was conducted on the relationship between P fractions, soil properties, and soil Olsen-P. This research emphasizes the significance of strategic phosphate fertilizer use in agriculture to ensure efficient production and to help address the global P scarcity.

**Keywords:** fertilizer; soil phosphorus; Olsen-P

## 1. Introduction

Phosphorus (P) is an essential element for plant growth and is involved in many important metabolic processes in plants, such as the synthesis of nucleic acids, the transport of proteins, etc. [1–3]; P fertilizer positively affects crop yields and is a critical component in ensuring national food security= [4]. In recent decades, the global P fertilizers consumption has increased dramatically. However, P remains one of the most important nutrients that limit the production of crops due to the low P use efficiency (18–20%) [5,6].

P fertilizer applied to soil is rapidly adsorbed by the soil and typically combines with $Al^{3+}$ and $Fe^{3+}$ in acidic soils and $Ca^{2+}$ in alkaline or neutral soils converted into non-labile P fractions [7,8]. Total phosphorus (TP) and Olsen-P are important indicators for evaluating the status of soil P pools [9,10]. However, the dynamic changes among different P fractions directly affect the effectiveness of soil P supply for crops [11,12]. There are different relationships between different soil P fractions and soil Olsen-P. Jimenez et al.

found that the Al-P and Fe-P have substantial effects on Olsen-P in albic soil [13]. Chen et al. also found a significant linear positive correlation between the Al–P and Olsen-P (R > 0.9) [14]. Mahmood et al. found that $NaHCO_3$-Po, $NaHCO_3$-Pi, NaOH-Pi, and HClD-Pi were significantly positively correlated with grain yields, respectively [15]. The soil P dynamics are influenced by abiotic (organic matter, metals, pH, etc.) and biotic (microorganisms, phosphatase, etc.) [16–18]. In addition, land use patterns and fertilization can also greatly affect the changes in P fractions. Luo et al. reported that the season, peatland type, and soil depths strongly affected P fractions in Zoige soil [19,20]. Yan et al. found that manure incorporation significantly increased P fractions of $H_2O$-Pi, $H_2O$-Po, $NaHCO_3$-Pi, and HCl-Pi [1]. The transformation between different P fractions can also affect the effectiveness of soil P.

Studies in other regions have shown that organic fertilizers and organic matter can enhance the activity and availability of phosphorus [21,22]. For example, pigeon dung tea promotes phosphorus availability and wheat growth in a calcareous sandy soil by decreasing the P adsorption [23]. Citric acid-modified biochar increased the concentration of the available phosphorus [24]. The effect of using diammonium phosphate in combination with other crop residues (sand pine and sesame) is better than using inorganic phosphorus fertilizer alone [25]. Under semi-humid conditions, the combined use of phosphorus fertilizer and farmyard manure enhances wheat productivity by improving the soil quality and phosphorus availability in calcareous soil [26].

The Loess Plateau is an important grain production base in China, accounting for about 56% of the nation's total land area [27]. The winter wheat–summer maize rotation is the most dominant cropping system in the region, and they are an indispensable part for ensuring food security in our country [28,29]. Excess P fertilizers have been applied to pursue high yields in recent decades. However, most P fertilizers are fixed by the soil in the northwest region of China due to the high pH, calcium carbonate, etc. [30,31]. This makes it difficult for P fertilizer to be efficiently used by plants, and a large amount of P accumulates in the soil. Many varieties of mineral P fertilizers are applied to the arable land [32]. The mismatch between P fertilizer varieties, soil, and crops is also one of the important reasons for the low P utilization rate and the deterioration of the soil quality [33].

Does the special ecological and climatic environment of the Loess Plateau affect the transformation of phosphate fertilizer differently from other areas? It is necessary to explore the transformation process of different P fertilizer varieties in the soil under the winter wheat–summer maize rotation system to ensure the development of sustainable agriculture in the northwest region and alleviate the current P rock resource shortage in China. Therefore, we conducted a four-year (2017–2021) field experiment to obtain valuable information on the transformation of different P fertilizers in the soil. The aim of this work includes the following: (i) exploring the transformation process of different P fertilizer varieties in the Loess Plateau, (iii) understanding the relationship between different P components and soil P availability, and (iii) proposing a reasonable P fertilizer application strategy.

## 2. Materials and Methods

### 2.1. Site Description

A four-year field experiment under a wheat and corn rotation system was conducted from October 2017 to September 2021. The experiment site was at the Experimental Station of Northwest Agricultural and Forestry University, in Yangling (34°17′44′ N, 108°04′10′ E, altitude of 520.3 m a.s.l.), Shaanxi, China (Figure 1). This region is situated in the dryland area of the southern edge of the Loess Plateau, soil classification is phaeozems (FAO 90). The properties of the soil (0–20 cm soil layer) are shown in Table 1. The mean monthly rainfall and temperature data at the experimental site are shown in Figure 2.

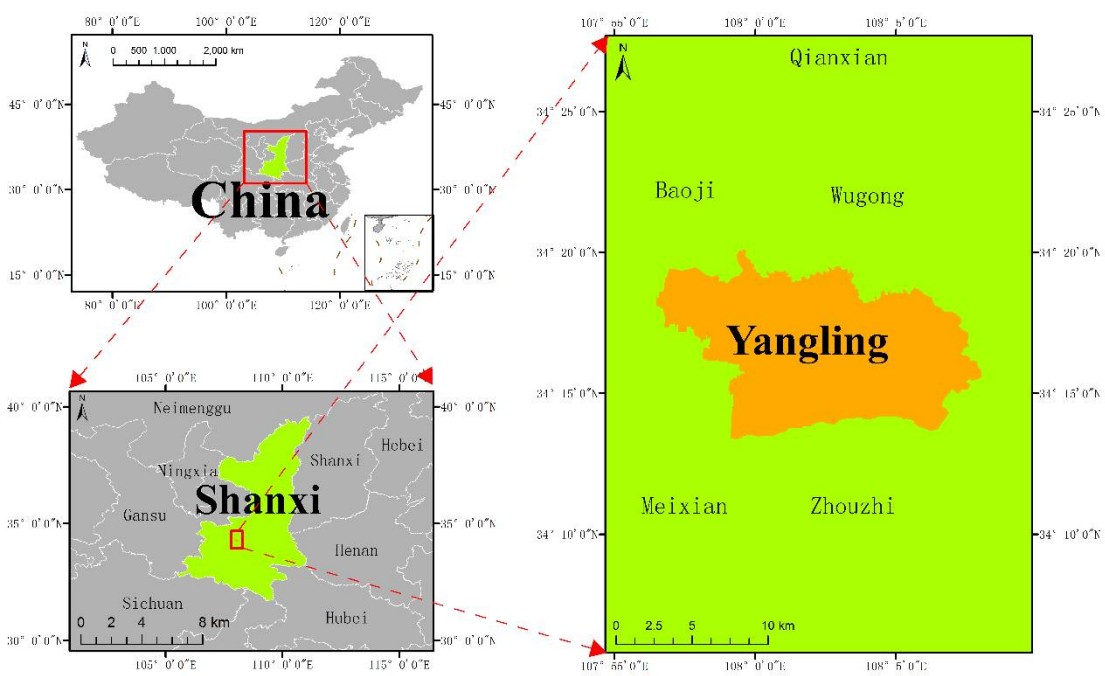

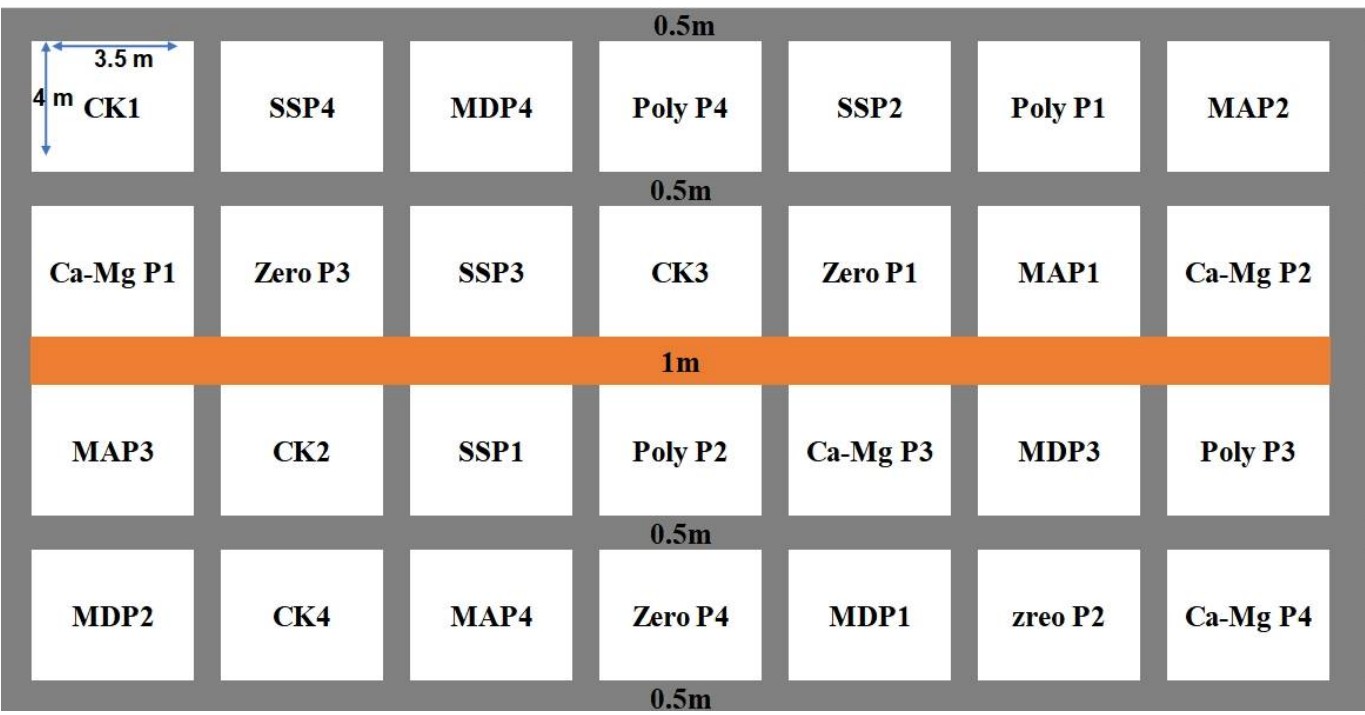

**Figure 1.** Location of the study area and plots distributed.

**Table 1.** Soil physical and chemical properties before the field experiment establishment.

| pH (1:2.5) | OC (g·kg⁻¹) | TN (g·kg⁻¹) | TP (g·kg⁻¹) | CaCO₃ (g·kg⁻¹) | AP (mg·kg⁻¹) | AK (mg·kg⁻¹) |
|---|---|---|---|---|---|---|
| 8.21 | 10.52 | 0.92 | 0.84 | 59.71 | 20.74 | 148.61 |

OC; organic carbon; TN: total nitrogen; TP: total phosphorus; AP: available phosphorous; AK: available potassium.

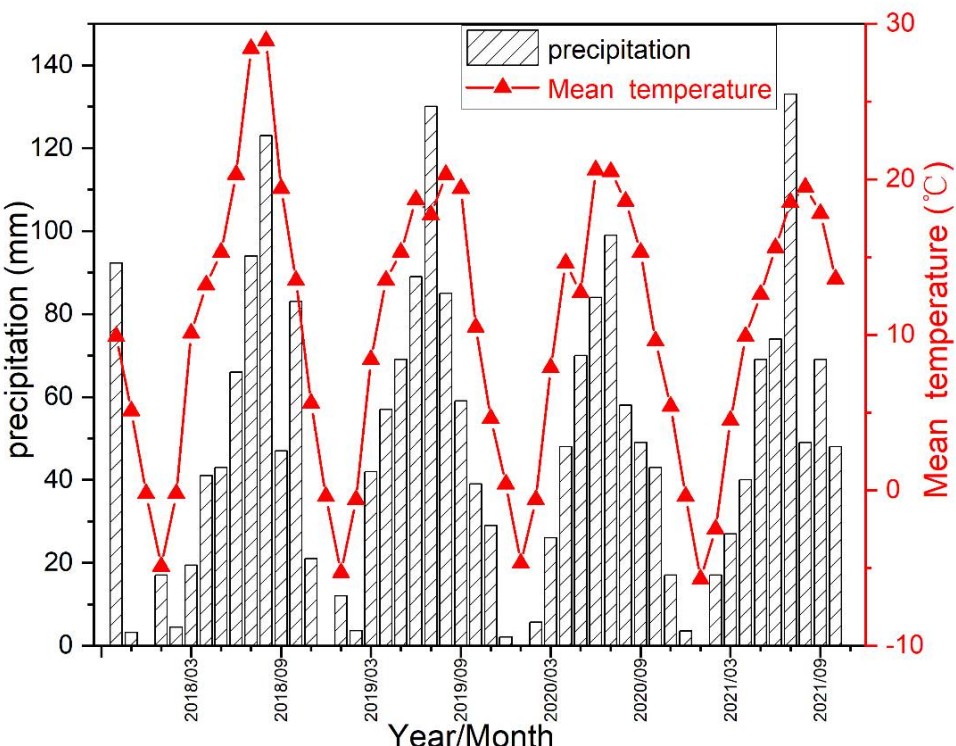

**Figure 2.** Monthly average temperature (°C) and distribution of precipitation (mm) during 2017–2021.

## 2.2. Experimental Design

Winter wheat (*Triticum aestivum* L.) was planted in mid-October and harvested in early June next year. Summer maize (*Zea mays* L.) was planted in mid-June and harvested in early October of the same year. Each experimental plot was 14 m² (3.5 m × 4 m) and arranged in a randomized block design with four replicates. The planting densities were 180 kg ha$^{-1}$ of seeds for wheat and 67,000 plants ha$^{-1}$ for maize, respectively. The wheat variety was "xiaoyan 22" and the maize variety was "zhengdan 958". Before planting, the basic fertilizers were spread to the experimental plot, and then the ground was plowed uniformly with a rotary tiller. Finally, the seeds were planted with a 25 cm-row plant spacing of wheat and 55 cm-row plant spacing of maize by a seeder. No irrigation was used, and the management of each plot was the same during the entire experimental period.

The experiment adopted a randomized block experimental design. There were seven treatments in the present experiment: CK (the control without any fertilizer), Zero P (only N and no P fertilizer), MAP (ammonium dihydrogen phosphate fertilizer, $(NH_4)H_2PO_4$, $P_2O_5 = 60.5\%$), MDP (diammonium phosphate fertilizer, $(NH_4)_2HPO_4$, $P_2O_5 = 53.8\%$), Ca-Mg P (calcium-magnesia phosphate fertilizer, $Ca_3(PO_4)_2 + CaSiO_3 + MgSiO_3$, $P_2O_5 = 18\%$), SSP (calcium superphosphate, $Ca(H_2PO_4)_2 \cdot H_2O$, $P_2O_5 = 45\%$), Poly P (ammonium polyphosphate, $(NH_4,H)_n + 2P_nO_{3n+1}$, $P_2O_5 = 58\%$). All fertilizers are produced by China National Chemical Corporation (Tianjin, China). According to local planting habits, P fertilizer was applied once as basal fertilizer at application rates of 115 kg $P_2O_5$ ha$^{-1}$ in wheat season and 90 kg $P_2O_5$ ha$^{-1}$ in maize season. Meanwhile, N application rates of wheat and maize were 120 kg N ha$^{-1}$ and 180 kg N ha$^{-1}$, respectively. N fertilizer (urea) was used as twice-split fertilization, 60% at pre-plant and 40% at the jointing stage of wheat or V12 (twelve leaf collar) maize [34]. According to local planting habits, potassium fertilizer is applied once every five years, and no potassium fertilizer was used during the experiment [35]. Each treatment was designed with four replicates. Crop yields are shown in Table 2.

**Table 2.** Grain yield in 2017–2020.

| Treatment | Wheat Grain Yield (kg ha$^{-1}$) | | | | | | |
|---|---|---|---|---|---|---|---|
| | 2017–2018 | 2018–2019 | 2019–2020 | 2020–2021 | Mean Yield (kg·ha$^{-1}$) | Change to CK (%) | Change to Zero P (%) |
| CK | 5537 d | 5081 e | 4840 d | 4296 e | 4939 | 0 | −18.3 |
| Zero P | 6249 c | 5956 d | 5745 c | 5415 d | 5841 | 18.3 | 0 |
| Poly P | 7995 a | 8029 ab | 8067 a | 8158 a | 8062 | 63.2 | 38.0 |
| MAP | 7913 a | 8164 a | 8017 a | 8097 ab | 8048 | 62.9 | 37.8 |
| Ca-Mg P | 6933 b | 7079 c | 7189 b | 7110 c | 7078 | 43.3 | 21.2 |
| SSP | 7811 a | 7840 b | 7898 a | 7901 b | 7863 | 59.2 | 34.6 |
| MDP | 6967 b | 7111 c | 7135 b | 7118 c | 7083 | 43.4 | 21.2 |
| Maize grain yield (kg ha$^{-1}$) | | | | | | | |
| | 2018 | 2019 | 2020 | 2021 | Mean yield (kg·ha$^{-1}$) | Change to CK (%) | Change to Zero P (%) |
| CK | 5045 e | 4825 d | 4665 d | 4431 e | 4742 | 0 | −16.1 |
| Zero P | 5840 d | 5678 c | 5482 c | 5028 d | 5507 | 16.1 | 0 |
| Poly P | 6295 b | 6311 a | 6459 a | 6548 a | 6403 | 35.0 | 16.3 |
| MAP | 6382 ab | 6368 a | 6398 a | 6432 a | 6395 | 34.9 | 16.1 |
| Ca-Mg P | 6052 c | 6081 b | 6005 b | 6071 c | 6052 | 27.6 | 9.9 |
| SSP | 6458 a | 6230 a | 6277 ab | 6269 b | 6309 | 33.0 | 14.6 |
| MDP | 6088 c | 5998 b | 6040 b | 6099 c | 6056 | 27.7 | 10.0 |

Note: CK: the control without any fertilizer; Zero P: only nitrogen and no phosphate fertilizer; SSP: calcium superphosphate; MAP: ammonium dihydrogen phosphate fertilizer; MDP: diammonium phosphate fertilizer; Ca-Mg P: calcium-magnesia phosphate fertilizer; Poly P: ammonium polyphosphate. Different lowercase letters indicate significant differences ($p > 0.05$).

*2.3. Sample Collection and Determination*

After the crop was harvested, soil samples (0–20 cm) were collected from 5 locations according to the "S" route in each experimental plot and then mixed. After air-drying, the soil samples were sieved through a ~2.0 mm screen and stored in sealed plastic jars for analysis. TP and Olsen-P were determined by the ammonium molybdate method described by Murphy and Riley [36]. Total nitrogen (TN) was determined by the Kjeldahl method [37]. Determination of soil organic carbon using potassium dichromate external heating penalty [38]. Soil pH was measured with a soil-to-distilled water ratio of 1:2.5.

At maturity, plants were harvested by cutting close to the ground. A part of the randomly selected plants was ground to pass through a 0.5 mm sieve and digested with concentrated sulfuric acid/hydrogen peroxide [39] to measure P content for the plant.

Soil P fraction contents were measured following the procedure modified by Tiessen and Sui et al. based on the method of Hedley et al. [40–42]. The detailed testing process (shown in Figure 3) was as follows: 0.5 g of soil (<100 mesh) was weighed into a 50 mL centrifuge tube and sequentially fractionated with 30 mL of deionized water, 0.5 M NaHCO$_3$, NaOH, and 1 M HCl, respectively. Each sample was shaken for 16 h at 200 oscillations min$^{-1}$ at 24 °C, followed by centrifugation for 12 h at 25,000 r/min under 0 °C. Pi concentration in the extracts filtered through a 0.45 mm cellulose membrane filter was measured using the ascorbic acid colorimetric method. Total NaHCO$_3$-P (Pt) and NaOH-P (Pt) concentrations were determined after digestion with ammonium persulfate and 0.9 M H$_2$SO$_4$. Po in the extracts was calculated as the difference between Pt and Pi. The residual-P was measured after soil digesting in concentrated H$_2$SO$_4$ (18 M) and H$_2$O$_2$. All reagents are produced by China National Pharmaceutical Group (Beijing, China).

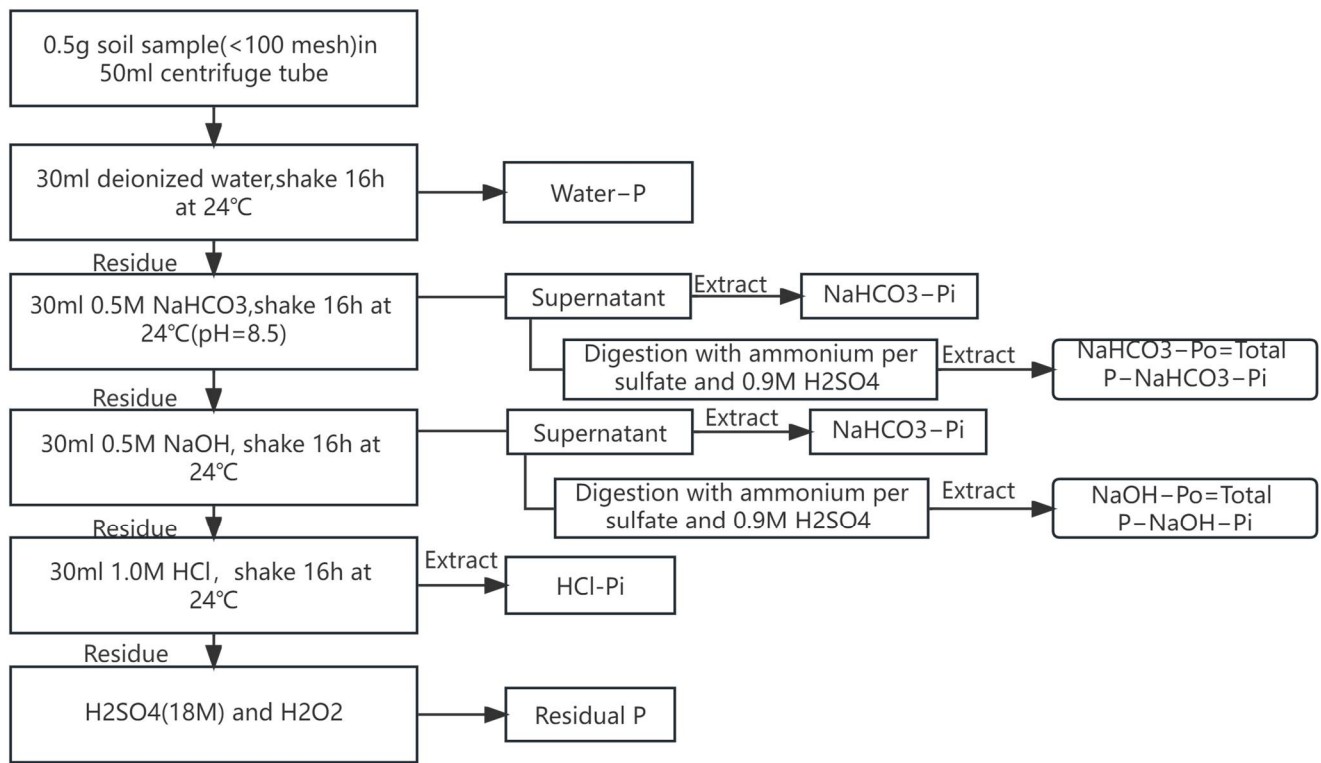

**Figure 3.** Flowcharts for soil inorganic P (Pi) fraction sequential extraction and soil organic P (Po) determination.

### 2.4. Data Calculation and Analysis

Crop P removal, P surplus and phosphorus of recovery efficiency (PRE) were calculated by the following formulas according to Devkota et al. and Lu et al. [43,44].

$$Crop\ P\ removal = Biomass \times P_{plant}$$

$$P\ surplus = F_P - U_P \tag{1}$$

$$PRE = \frac{(U_P - U_0)}{F_P} \tag{2}$$

where $P_{plant}$ is plant phosphorus content, $U_P$ and $U_0$ are aboveground crop P uptake in P fertilizer treatment plots and Zero P treatment plots, respectively. $F_P$ is the applied P fertilizer amount.

All the experiments were conducted four times, and the average values were reported. Normality and homoscedasticity tests performed, Analysis of variance among treatments and mean separation tests (Duncan's multiple range test and least significant difference test) were performed using the IBM SPSS statistics 24. The differences among means and correlation coefficients were considered significant when $p < 0.05$. All figures were generated using Origin 2021. Correlations between soil P composition, soil properties, and nutrient uptake by crops under different treatments were analyzed by the "corrplot" package in R (version 3.6.3). Principal component analysis (PCA) between soil P composition, soil properties, and nutrient uptake by crops were analyzed by Canoco 5.

## 3. Results and Discussions

### 3.1. P Surplus and PRE

The absorption of P by crops was affected by P fertilizer varieties (Table 3). Total P uptake of maize and wheat under CK, Zero P, SSP, MAP, MDP, Ca-Mg P, and Poly P treatment was 83.4, 113.4, 147.4, 150.2, 132.5, 134.2, and 162.3 kg P ha$^{-1}$ year$^{-1}$, respectively, during 2017–2021. Soil P decreased yearly for CK and Zero P treatment, while soil P content increased under other P application treatments (Table 3). Net P surplus in different treatments was 42.7–72.5 kg P ha$^{-1}$ year$^{-1}$. PRE of different treatments using different P fertilizer varieties was 9.17–23.85%. P uptake of crops was inversely correlated with soil P surplus.

**Table 3.** Mean annual N and P input, crop P removal, phosphate recovery efficiency, and P surplus in the different treatments from 2017 to 2021.

| Treatment | P Input | Crop P Removal | P Surplus | PRE (%) |
|---|---|---|---|---|
| | (Kg P ha$^{-1}$ Year$^{-1}$) | (Kg P ha$^{-1}$ Year$^{-1}$) | (Kg P ha$^{-1}$ Year$^{-1}$) | |
| CK | 0 | 83.4 e [1] | −83.4 e | - |
| Zero P | 0 | 113.4 d | −113.4 d | - |
| SSP | 205 | 147.4 b | 57.6 b | 16.59 b |
| MAP | 205 | 150.2 ab | 54.8 ab | 17.95 b |
| MDP | 205 | 132.5 c | 72.5 c | 9.17 c |
| Ca-Mg P | 205 | 134.2 c | 70.8 c | 10.24 c |
| Poly P | 205 | 162.3 a | 42.7 a | 23.85 a |

[1] Means followed by similar letters within each column were not significantly different ($p > 0.05$) based on analyses by one-way ANOVAs followed by Duncan multiple range tests. CK: the control without any fertilizer; Zero P: only nitrogen and no phosphate fertilizer; SSP: calcium superphosphate; MAP: ammonium dihydrogen phosphate fertilizer; MDP: diammonium phosphate fertilizer; Ca-Mg P: calcium-magnesia phosphate fertilizer; Poly P: ammonium polyphosphate. PRE: Phosphorus of recovery efficiency. Values statistics are based on four repeating treatments instead of based on the year.

In this work, the total accumulation of P in the soil was 170.8–290 kg P ha$^{-1}$ for four years. Previous studies have shown that soil P accumulation is greatly related to the amount of P fertilizer input and the P fertilizer varieties [1,45–47]. There was a significant difference in PRE of different P varieties (Table 3). The PRE in this study was lower than the reported range by Khan et al. and Syers et al. [31,47].This may be due to the generally low conversion efficiency of P fertilizers in soils with low fertility [48]. In this study, the PRE of the Poly P treatment was higher than that of others (Table 3), probably because there were competing adsorption sites and complexation sites between polyphosphate and phosphate, which reduced the adsorption of orthophosphates in the soil, thus making PRE higher. In addition, PRE under SSP and MAP treatments were also relatively high. Previous reports point out that the pH of P fertilizer can affect PRE [49]. The acidic substances carried by SSP and MAP can promote P dissolution in calcareous soils.

### 3.2. Soil Properties

The soil physicochemical properties were determined after the field experiment was conducted for 4 years (Table 3). The soil physicochemical properties showed obvious differences under different treatments. The soil pH varied from 7.83 to 8.23. Soil organic C showed a slight downward trend, possibly due to the lack of carbon replenishment. However, there were no significant differences among organic C of different treatments. The application of N fertilizers increased TN content (Table 4). TN contents under CK, Zero P, SSP, MAP, MDP, Ca-Mg P, and Poly P treatments were 0.68, 0.84, 0.89, 0.95, 0.93, 0.76, and 1.03 g kg$^{-1}$, respectively. The content of $CaCO_3$ showed no significant difference among different treatments, and the value varied from 59.01 to 59.92 g kg$^{-1}$. There was also no significant difference in available K content (146.32–148.12 mg·kg$^{-1}$) in different treatments. Crop productivity is closely related to soil physicochemical properties [50]. In this study, fertilization slightly reduced the soil pH, and similar results were seen in the report by Hati et al. [51].

**Table 4.** Soil (0–20 cm topsoil) physical and chemical properties under different fertilization treatments in the field after crop harvest in October 2021.

| Treatment | pH (1:2.5) | Organic C (g·kg⁻¹) | Total N (g·kg⁻¹) | Total P (g·kg⁻¹) | CaCO₃ (g·kg⁻¹) | Available P (mg·kg⁻¹) | Available K (mg·kg⁻¹) |
|---|---|---|---|---|---|---|---|
| CK | 8.23 a [1] | 9.83 b | 0.68 c | 0.73 c | 59.3 a | 18.17 c | 148.11 a |
| Zero P | 8.19 a | 9.92 b | 0.84 b | 0.69 c | 59.21 a | 17.03 c | 148.12 a |
| SSP | 8.13 ab | 10.04 a | 0.89 ab | 0.85 b | 59.12 a | 21.03 b | 147.25 a |
| MAP | 7.93 c | 10.08 a | 0.95 a | 0.86 ab | 59.16 a | 22.03 b | 146.32 a |
| MDP | 7.83 c | 9.91 ab | 0.93 a | 0.88 a | 59.11 a | 21.45 b | 147.42 a |
| Ca-Mg P | 8.11 b | 10.01 a | 0.76 c | 0.91 a | 59.92 a | 22.35 ab | 146.32 a |
| Poly P | 8.01 c | 10.13 a | 1.03 a | 0.89 a | 59.01 a | 23.56 a | 146.93 a |

[1] Means followed by similar letters within each column were not significantly different ($p > 0.05$) based on analyses by one-way ANOVAs followed by Duncan multiple range tests. CK: The control without any fertilizer; Zero P: only N and no P fertilizer; SSP: calcium superphosphate; MAP: ammonium dihydrogen phosphate fertilizer; MDP: diammonium phosphate fertilizer; Ca-Mg P: calcium-magnesia phosphate fertilizer; Poly P: ammonium polyphosphate.

### 3.3. TP and Olsen-P

Olsen-P and TP content were closely related to P fertilizer varieties (Table 3 and Figure 4). Olsen-P and TP contents were decreased year-by-year for CK and Zero P treatment during 2017–2021 due to the plants absorbing part of soil P. The TP contents under CK, Zero P, SSP, MAP, MDP, Ca-Mg P, and Poly P treatments were 0.68, 0.84, 0.89, 0.95, 0.93, 0.76, and 1.03 g kg⁻¹, respectively. The corresponding content of Olsen-P was 18.17, 17.03, 21.03, 22.03, 21.45, 22.35, and 23.56 mg kg⁻¹, respectively. The TP contents of topsoil (0–20 cm) under SSP, MAP, MDP, and Poly P treatments were all significantly higher than that of the Ca-Mg P treatment. Olsen-P of Ca-Mg P and Poly P treatment were higher than others.

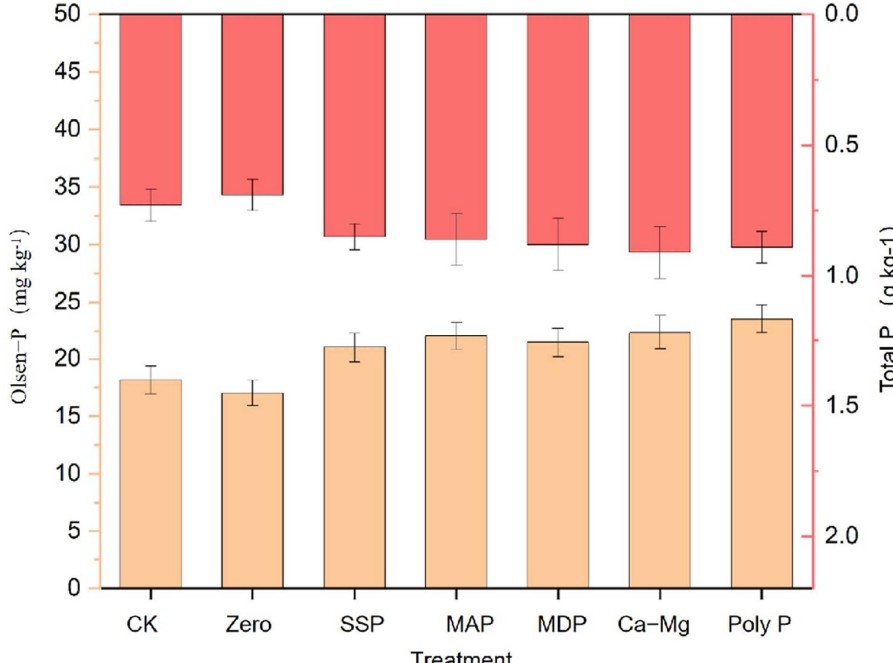

**Figure 4.** Total P (red) and Olsen-P (orange) under different fertilization treatments in the field after crop harvest in Oct. 2021. CK: the control without any fertilizer; Zero P: only nitrogen and no phosphate fertilizer; SSP: calcium superphosphate; MAP: ammonium dihydrogen phosphate fertilizer; MDP: diammonium phosphate fertilizer; Ca-Mg P: calcium-magnesia phosphate fertilizer; Poly P: ammonium polyphosphate.

The Olsen-P contents of the P fertilizer application plots were 22.1–25.3 mg kg$^{-1}$ for the wheat season during 2017–2021, while 20.6–24.3 mg kg$^{-1}$ for the maize season during 2018–2021. Previous studies have also confirmed that long-term fertilization can increase the soil TP and Olsen-P [52,53]. The Olsen-P content of the P fertilizer application was increased by 15.7–29.7% compared with the Zero P treatment. The TP content of the P fertilizer application was increased by 16.4–24.7% compared with the Zero P treatment (Figure 4). The Olsen-P under Poly P treatment was the highest because the release of P in Poly P may be slow.

### 3.4. Soil P Fractions

HCl-Pi content was the highest among all the phosphorus components (422.6–532.3 mg kg$^{-1}$) (Table 5). The approximate order of soil P fractions content was the following: HCl-Pi > residual-P > NaOH-Pi > NaOH-Po > NaHCO$_3$-Pi > NaHCO$_3$-Po ≈ water-Pi (Table 5). The effects of different P fertilizer varieties on the soil P fractions differed. The water-Pi of the SSP treatment was significantly lower than that of other P treatments. The contents of NaHCO$_3$-Pi, NaHCO$_3$-Po, NaOH-Pi, NaOH-Po, and Residual-P for different treatments were varying from 22.2 to 39.1 mg kg$^{-1}$, 13.1 to 29.2 mg kg$^{-1}$, 84.2 to 115.8 mg kg$^{-1}$, 55.4 to 71.7 mg kg$^{-1}$, and 106.6 to 158.7 mg kg$^{-1}$, respectively. The P fractions of Poly P treatment were relatively high, which also coincided with the highest TP content in the Poly P treatment. The contents of water-Pi, NaOH-Pi, and HCl-Pi under SSP treatment were all relatively low. The content of NaOH-Po under MAP treatment was relatively low. The NaHCO$_3$-Pi under Ca-Mg P treatment was relatively low. The content of NaHCO$_3$-Po, NaOH-Pi, and NaOH-Po under MDP treatment was relatively low.

**Table 5.** Concentrations (mg kg$^{-1}$) of different P fractions under different fertilization treatments.

| Treatment | Water-Pi | NaHCO$_3$-Pi | NaHCO$_3$-Po | NaOH-Pi | NaOH-Po | HCl-Pi | Residual-P | Sum of Pi | Sum of Po |
|---|---|---|---|---|---|---|---|---|---|
| CK | 5.1 c [1] | 22.2 c | 13.1 b | 86.2 c | 55.4 c | 429.7 c | 106.6 b | 541.2 c | 68.5 b |
| Zero P | 7.4 c | 25.6 c | 17.2 b | 84.2 c | 57.1 b | 422.6 c | 111.4 b | 539.8 c | 74.3 b |
| SSP | 22.2 b | 33.7 ab | 28.4 a | 94.6 b | 69.5 a | 503.2 b | 149.3 a | 653.7 b | 97.9 a |
| MAP | 28.5 a | 37.8 a | 25.6 a | 104.6 a | 64.2 b | 532.3 a | 153.2 a | 703.2 b | 89.8 a |
| Poly P | 29.2 a | 39.1 a | 29.2 a | 115.8 a | 71.7 a | 521.2 a | 155.3 a | 705.3 b | 100.9 a |
| Ca-Mg P | 28.5 a | 32.5 b | 25.5 a | 102.2 a | 68.8 a | 521.2 a | 158.7 a | 684.4 a | 96.3 a |
| MDP | 27.2 a | 34.3 ab | 23.6 b | 95.2 b | 61.2 b | 525.8 a | 156.8 a | 682.5 a | 80.8 a |

[1] Means followed by similar letters within each column were not significantly different ($p > 0.05$) based on analyses by one-way ANOVAs followed by Duncan multiple range tests. CK: the control without any fertilizer; Zero P: only nitrogen and no phosphate fertilizer; SSP: calcium superphosphate; MAP: ammonium dihydrogen phosphate fertilizer; MDP: diammonium phosphate fertilizer; Ca-Mg P: calcium-magnesia phosphate fertilizer; Poly P: ammonium polyphosphate. Note: Sum of Pi = Water-Pi + NaHCO$_3$-Pi + NaOH-Pi + HCl-Pi; Sum of Po = NaHCO$_3$-Po + NaOH-Po.

Previous studies have shown that the transformation process of different P fertilizers in the soil differed [54,55]. The content of P fractions in soil is affected by soil properties, microbial activity, and fertilization [56,57].

In this study, P fertilizer increased the uptake of P by crops due to P fertilizers significantly increased soil P fractions, particularly the labile P fractions (Tables 2 and 4). P fertilizers can increase the soil soluble Pi content [52,58]. Many long-term field experiments have shown that P fertilization application can significantly increase crop yield and P uptake [59,60].

### 3.5. Relationships between Olsen-P and P Fractions

Olsen-P is one of the most important indicators for evaluating soil quality because it can be absorbed by plants easily. There was a significant positive linear correlation ($R^2 > 0.861$) among all P fractions and between the various P fractions and Olsen-P (Figures 5 and 6A). Water-Pi, NaHCO$_3$-Pi, NaHCO$_3$-Po, NaOH-Pi, NaOH-Po, HCl-Pi, and Residual-P were significantly positively correlated ($R^2$ = 0.957, 0.918, 0.876, 0.932, 0.861, 0.891, and 0.872, respectively) with Olsen-P (Figure 5). Some P components cannot

be directly absorbed by plants, so the contribution of various P fractions to Olsen-P cannot be directly reflected according to their correlation coefficient [61].

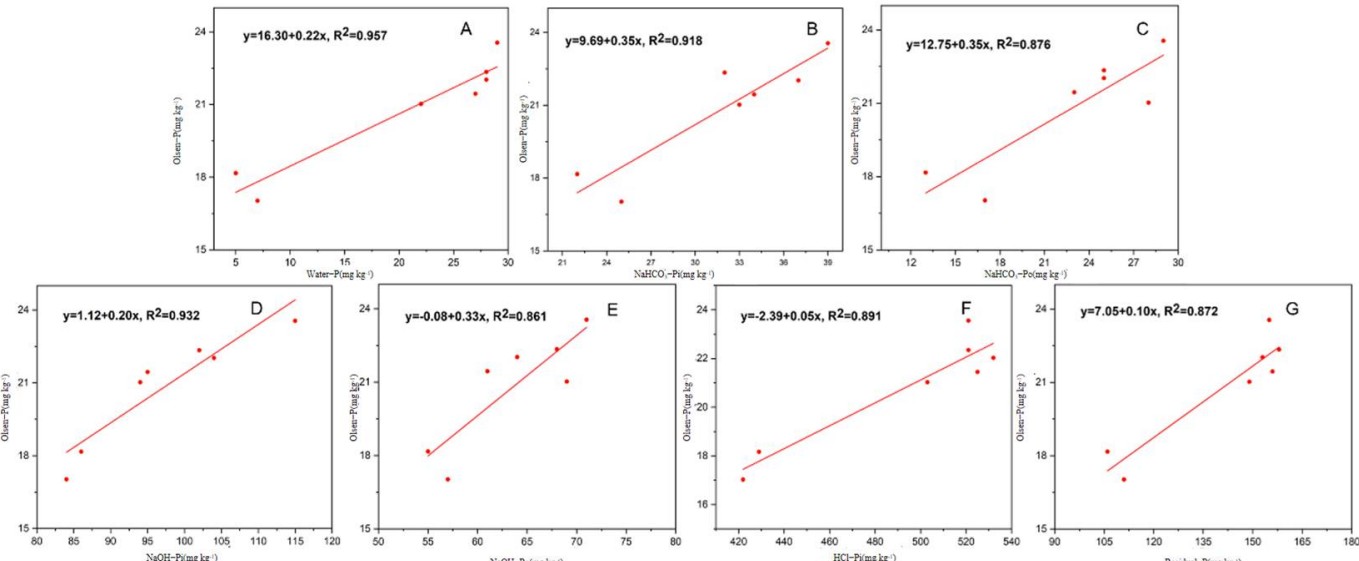

**Figure 5.** Relationship between Olsen-P and P fractions. (**A**–**G**) represents respectively the correlation about Olsen−P with Water−P (**A**), NaHCO$_3$−Pi (**B**), NaHCO$_3$−Po (**C**), NaOH−Pi (**D**), NaOH−Po (**E**), HCl−Pi (**F**) and Residual−P (**G**).

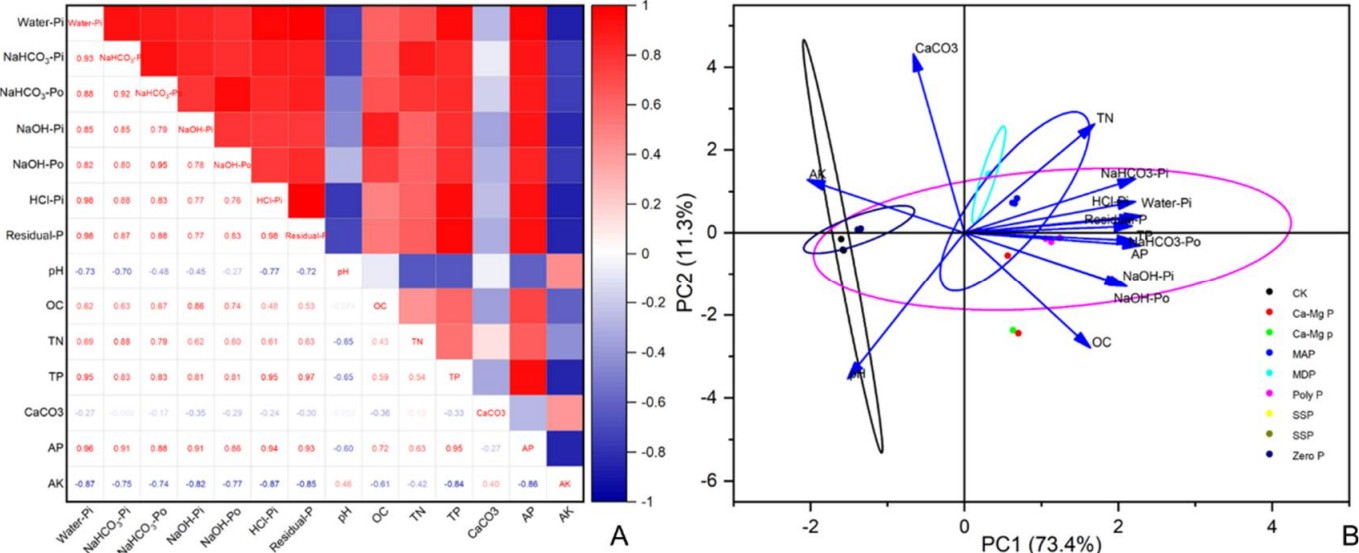

**Figure 6.** Pearson correlation coefficients (**A**) and principal components analysis (**B**) between soil properties and P fractions. CK: the control without any fertilizer; Zero P: only nitrogen and no phosphate fertilizer; SSP: calcium superphosphate; MAP: ammonium dihydrogen phosphate fertilizer; MDP: diammonium phosphate fertilizer; Ca−Mg P: calcium-magnesia phosphate fertilizer; Poly P: ammonium polyphosphate. OC; organic carbon; TN: total nitrogen; TP: total phosphorus; AP: available phosphorous; AK: available potassium.

To further understand the P transformation process, the relationships among P fractions were analyzed by path analysis (Table 6). The direct path coefficients of water-Pi, NaHCO$_3$-Pi, NaHCO$_3$-Po, NaOH-Pi, NaOH-Po, HCl-Pi, and Residual-P for Olsen were 0.782, 0.74, 0.132, 0.267, 0.227, −0.25, and −0.853, respectively. The indirect and direct path coefficients of Water-Pi on Olsen were the highest (Table 6). This result was consistent with

the study of Li et al. [62]. This shows that the metabolic process of phosphorus in the study area is consistent with other people's studies [63].

**Table 6.** Direct and indirect path coefficients of soil phosphorus (P) fractions on Olsen-P.

| Variable | Direct Path Coefficient | Indirect Path Coefficient | | | | | | |
|---|---|---|---|---|---|---|---|---|
| | | Water-Pi | NaHCO$_3$-Pi | NaHCO$_3$-Po | NaOH-Pi | NaOH-Po | HCl-Pi | Residual-P |
| Water-Pi | 0.782 | | 0.687 | 0.116 | 0.226 | 0.185 | −0.245 | −0.838 |
| NaHCO$_3$-Pi | 0.74 | 0.618 | | 0.122 | 0.226 | 0.182 | −0.22 | −0.746 |
| NaHCO$_3$-Po | 0.132 | 0.692 | 0.684 | | 0.211 | 0.216 | −0.207 | −0.747 |
| NaOH-Pi | 0.267 | 0.662 | 0.626 | 0.104 | | 0.177 | −0.176 | −0.656 |
| NaOH-Po | 0.227 | 0.638 | 0.594 | 0.125 | 0.209 | | −0.191 | −0.704 |
| HCl-Pi | −0.25 | 0.766 | 0.650 | 0.109 | 0.207 | 0.173 | | −0.838 |
| Residual-P | −0.853 | 0.769 | 0.647 | 0.116 | 0.205 | 0.187 | −0.246 | |

*3.6. Relationships between P Fractions and Soil Properties*

To reveal the relationship between P fractions and soil properties, a Pearson correlation of soil properties and P fractions was analyzed (Figure 6A). The P fractions showed a significantly positive correlation between TP and Olsen-P (Figure 6A). This is consistent with other studies on other soils [11,15,64]; however, the soil pH and Olsen-P had a negative correlation. The pH may affect the conversion process of P fertilizers in the soil and the conversion among different P fractions [65]. The available K and soil CaCO$_3$ also showed a negative correlation with P fractions, respectively. CaCO$_3$ is closely related to pH [66], and too much active phosphorus will destroy the balance of phosphorus and potassium, causing potassium deficiency [67].

The results of PCA also showed the relationship between P fractions and soil properties (Figure 6B). PC1 and PC2 explained 73.4% and 11.3% of the total variance, respectively. PC1 can be interpreted as a difference in properties caused by different varieties of P fertilizers. PC1 made a clear distinction between CK, Zero P, and other P treatments. As can be seen from the PCA, mineral fertilizers increased the soil N and P content and reduced the soil pH. The soil pH in the study area is relatively high (>7.5), so reducing the pH with mineral fertilizers can help improve the quality of farmland [68].

## 4. Conclusions

P fertilizer application had obvious effects on soil properties, P pools, and P fractions. The transformation process of different P fertilizer varieties in the soil was also different. TP and Olsen-P were supplemented due to the application of P fertilizer, while the TP and Olsen-P content in no P treatment was decreased yearly. The content of each P fraction was also higher than that of CK and Zero P treatment. Findings from long-term experiments showed that Poly P (ammonium polyphosphate), SSP (calcium superphosphate), and MAP (ammonium dihydrogen phosphate fertilizer) were more suitable for application in the Loess soil than MDP (diammonium phosphate fertilizer) and Ca-Mg P (calcium-magnesia phosphate fertilizer). The use of phosphate fertilizer helps reduce the pH of alkaline soil and improve soil quality.

**Author Contributions:** Writing—original draft preparation, C.L.; validation, X.L.; methodology, L.F.; software; formal analysis, writing—review and editing N.J.; investigation, Q.Y.; resources, data curation, J.L. All authors have read and agreed to the published version of the manuscript.

**Funding:** This research was funded by the Key special projects of the Ministry of Science and Technology for the 13th Five Year Plan of China (Project No. 2017YFD0200205) and the Project of Shaanxi Agricultural Science and Technology Innovation Drive (NYKJ-2020-YL-21; NYKJ-2021-YL(XN)19). Natural Science Foundation of China (No.41961124006).

**Institutional Review Board Statement:** Not applicable.

**Informed Consent Statement:** Not applicable.

**Data Availability Statement:** Data are contained within the article.

**Conflicts of Interest:** The authors declare that they have no known competing financial interests or personal relationships that could have appeared to influence the work reported in this paper.

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
