# Peer review of "Optimizing Phosphorus Fertilizer Use on the Loess Plateau: Impact on Soil Properties and Crop Production Efficiency"

_soilsystems, doi:10.3390/soilsystems8010003_

Round 1

Reviewer 1 Report

Comments and Suggestions for Authors

Dear authors congratulations on your work,

Please check below some points to be improved before publishing it.

Kind regards.

Abstract

-Experimental design must be added.

Introduction

-        Add a short review about the role of the biofertilizers and soil organic matter on P availability, adsorption, and availability. At least 5-10 references.

-        Add the hypothesis tested during this study. Suggest a separate paragraph with the hypothesis, followed by the main objectives.

Material and methods

-        Add soil classification according to the international standards (FAO).

-        Add the soil texture and classification of the experimental area.

-        Randomized Complete Block Design (RCBD)

-        Add scientific names of maize and wheat

-        Lines 111-120. Add the molecular formula of each fertilizer. Keep the full name but add the formula too.

-        Add some supplemental figures showing the plots with maize and wheat, also, if you have how the fertilizer was added.

-        Check the verbal time/rhythm of the sentences (e.g., “Each samples need to be shaken…” line 136), and check all these time issues throughout the article.  

-        Add the normality and homoscedasticity tests performed.  

Results

-        I suggest adding the data regarding the DM productivity of each crop. Add to the material and methods section.

-        Did you analyze the content of P in each crop? If so, add to the material and methods section too.

-        How did you reach the actual removal of P from the soil?

-        Table 1. Add a reference that PRE means Phosphate recovery efficiency

The discussion section is missing! Add it or combine it with the results. Check the guidelines of Soil system.

Conclusions

-        Avoid abbreviations in the conclusion add the full names of treatments etc..

English looks okay. However, double-check it using the word add on Grammarly. I found that several “the” were not necessary to be added (e.g., at the beginning of the conclusion “The P fertilizer”….). 

Comments on the Quality of English Language

Check the article using Grammarly too.

Reviewer 2 Report

Comments and Suggestions for Authors

Introduction

Line 44 is „Al3+ and Fe3+ in acidic soils and Ca2+„ shoud be „Al3+ and Fe3+ in acidic soils and Ca2+Please correct the way of writing throughout the manuscript. Please use subscript and superscript.

Line 53 and 60 Please correct the way of writing throughout the manuscript.

Material and Methods

Line 89 Authors should give information what type of soil use in experiment using WRB 2015 classification

Line 116-117 Authors should explain why the doses of phosphorus were so high.

Line 119 Authors should explain why don’t applicate potassium fertilizer in experiment?

Line 122 Authors should indicate from what depth they took soil samples.

Line 164 Results please change name of this chapter from Results witch Results and Discussion

Authors should necessarily provide the yields of wheat and corn.

Lines 213 The authors should necessarily provide in the Material and Method chapter how the content of Organic C and Total N in the soil was determined

The discussion is very poor. Two publications were used in Chapter 3.2, in Chapter 3.3. two publications, in chapter 3.4. seven publications, two publications in chapter 3.5, and in chapter 3.6 the obtained results were not discussed at all. General note. Authors should rebuild and significantly improve the discussion of the results.

The References chapter must be revised in accordance with the guidelines for authors

Reviewer 3 Report

Comments and Suggestions for Authors

Round 2

Reviewer 2 Report

Comments and Suggestions for Authors

The authors made significant corrections. The manuscript may be published in present form.

Reviewer 3 Report

Comments and Suggestions for Authors

Dear authors 

The revised version has been provided now. It was revised accordingly and improved as directed.

Thanks for providing this new version the all the scientific community.